# The Growth of Mountain Tourism in a Traditional Forest Area of Greece

**Stilianos Tampakis \*** , **Veronika Andrea, Paraskevi Karanikola** and **Ioannis Pailas**

Department of Forestry and Management of the Environment and Natural Resources, School of Agriculture and Forestry, Democritus University of Thrace, 193 Pantazidou St., 68200 Orestiada, Greece; vandrea@fmenr.duth.gr (V.A.); pkaranik@fmenr.duth.gr (P.K.); johnpailas879@hotmsil.com (I.P.)

\* Correspondence: stampaki@fmenr.duth.gr; Tel.: +30-25520-41134

**Abstract:** The recreational value of forests in mountain areas creates significant potential for local growth. Indeed, in recent decades, it has been noted that there has been an increase in the popularity of forests recognized as tourism destinations with a strong recreational importance. In the forest area of Metsovo, the locals are aware of the role of tourism in local growth, and it is considered, along with forestry and livestock farming, as the major critical advantage for sustainable development. Research Highlights: Although the locals are satisfied with the quality of their lives, they consider that mountain tourism should be enhanced with certain prerequisites, such as forest infrastructure and improvement of the road network. Background and Objectives: The case study aims to examine and interpret the locals' views in the mountain area of Metsovo on different factors that are able to support and encourage the growth of mountain tourism. Materials and Methods: Simple random sampling was applied, and data collection took place in 2018. In order to analyze and synthesize the locals' views, reliability, factor, and hierarchical cluster analyses were used. Results: The main findings of the survey indicate that according to the locals' views, there is a need for strategic organization addressing primarily forest recreation infrastructures from the Forest Service. Conclusions: The locals' views are focused on mountain tourism-related exploitation with the aim of forest recreation infrastructures. Indeed, the locals acknowledge the important role of the forest service in conservation schemes, but they also identify that there are forest recreational potentials in their area that need to be enhanced by the central administration and locally by the Forest Service.

**Keywords:** mountain tourism; forest recreation; locals views; Metsovo; infrastructures; forest policy

## 1. Introduction

Tourism is considered to produce multidisciplinary economic activity [1]. It is the largest global industry in terms of both employment and investments, highlighting its importance to growth at local, regional, national, and international level [2]. With regard to the current trends in tourism demand, it is acknowledged that there has been a growing interest in nature-based and mountain tourism [3], especially during the last decade [4]. Furthermore, nature-based and mountain tourism have been identified as the major pillar for sustainable development in rural and mountainous areas, while in regions with natural assets, prospects for new economic opportunities arise along with environmental protection [5].

Mountainous forest areas cover almost 28% of the Earth's surface [6], while it is acknowledged that conservation and sustainable management in these areas are two closely affiliated concepts. Indeed, as Nischalke et al. [7] highlighted, coffee production in forests in Southwestern Ethiopia has engaged with cultural assets of the local community, enhancing their positive attitude towards forest

conservation. In the same concept, Toscani and Sekot [8] presented the critical role of small-scale forestry in the Austrian Alps.

An international shift towards mountain tourism demand has been recorded in the ecotourism industry, as alpine regions offer an impressive arsenal of tourist resources, such as spectacular landscapes of relief impresses through altitude, stunning cliffs, scenic ridges, and a variety of genetic types of relief; distinct activities in ski areas; observation of rare flora and fauna species; special areas ideal for other winter sports; hunting; educational and scientific purposes; fishing; etc. [9]. Mountain tourism encompasses a range of hard-level activities, such as rock climbing, long and challenging treks, and mountain biking, as well as soft activities, including hiking and low-intensity treks. Whilst both kind of activities require proper landscape planning, with a good quality of infrastructure and accessibilities, they are able to provide comfortable experiences [10].

In Greece, mountainous areas account for 77.9% of the total country area, and semimountainous municipalities cover 71.3% [11]. From early on, forest policy for the mountainous areas of Greece was aimed at protecting the environment and supporting local populations. In particular, the Presidential Decree No. 19-11-1928 (Government Gazette 252/30-11-1928/T.A') states in Article 1 that, "Forest management, whether the forests are public or not, regulates the position, extend, amount, type, and time of logging, as well as the total of the forest economy and its yields to the fullest extent possible for the purpose of the forest economy and, to the extent possible, to meet social needs of the country specifically addressing to the host local community"; while according to Article 2, "The management and exploitation of forests, public and private, are distinguished by sustainable or continuous exploitation and periodic or intermittent use". It is therefore acknowledged that in mountainous areas, the protection and preservation of the forest and natural environment is focused on locals' well-being in line with sustainability principles.

Thus, it is of utmost importance for decision makers and managers to prioritize destination development goals, such as improvement of the quality of sustainable tourism elements, core competences, environmental protection, infrastructural establishment, and maintenance, addressing tourism product attractiveness, and service quality. Therefore, it is vital to identify the social representations of sustainable mountain tourism [12]. Along the same line, the Recreation Opportunity Spectrum planning system has been used as the roadmap for planning and improving various recreational opportunities in natural areas. This system comprises a valuable tool in designing the proper model to meet both environmental preferences and traditional forest management, while it also takes into account existing opportunities and accessibility issues [13].

Corresponding to Tampekis et al. [14], who examined an optimum evaluation method for the forest road networks in Metsovo, it has been underlined that there is a need for an integrated strategy in planning and upgrading transportation schemes in mountainous areas. This method should be based on the diverse character of sustainable development and address eco-efficient nodes and bioeconomies for the exploitation of the natural resources. The same authors also stress that it is significant to motivate human and social resources towards the promotion of cultural assets and the special financial features of the case area.

In several studies, tourists' views were recorded in terms of their level of satisfaction and preferences [4,15,16] in order to evaluate the dual aim of conservation and recreation, whereas in this study, the locals' views were examined according to recent research findings that list local people as a special interest group, which assesses tourism infrastructures in a more severe way compared to tourists, as these infrastructures primarily affect the quality of their lives on a daily basis [5,17–20]. In particular, it is acknowledged that the creation and improvement of tourism infrastructures has always been regarded as a useful policy tool aimed at rural development and tourism growth [21]. Added to that and taking into consideration the increasing mountain and nature-based tourism demand and tradeoffs, it is obvious that there is also a need for an improvement and redevelopment of sustainable facilities and infrastructures in mountainous areas.

In addition to the above, we must also mention that mountain tourism, which is closely affiliated with forest recreation, should provide access to proper accommodations and amenities aiming to meet the needs of natural recreation seekers [22] but also of local people. On the other hand, it should also be noted that in areas of high naturalness, there are difficulties in accessibility, ascribed to poor transportation systems, while human demands for natural resources continue to grow rapidly [23].

To this end, the traditional and alternative values of the forest providing various management objectives should address the integration of all its functions, including timber, biodiversity, protection from soil erosion and floods, and carbon sink, as well as its recreational services [24]. Recent findings revealed the positive effects of forest recreational activities in mental health and psychological and physiological wellbeing ("forest bathing") [25,26], which declares the need for the engagement and further development of forest recreation schemes and infrastructures. Tourism growth is highly dependent on infrastructures, such as the road network, transportation, accessibility amenities, and hotels, provided in the destination, while it is also appraised that infrastructural development influences local people in a positive way towards tourism development [27].

The aim of the study was to examine and interpret the locals' views in the mountain area of Metsovo, one of the most alpine regions in Greece, situated in the northwest part of the country, regarding the evaluation of different factors that are able to support and encourage the growth of mountain tourism. According to the main findings reflecting the locals' views, the need was recognized for further infrastructural development addressing mountain tourism at the forest area in Metsovo, with a special focus on forest recreation infrastructures. Traditional forest management and its services were positively evaluated in terms of efficiency, while the forest recreational potential was considered to be lagging on account of the central administration and, locally, the Forest Service.

## 2. Materials and Methods

### 2.1. Study Area

The Municipality of Metsovo (Figure 1) administratively belongs to the Prefecture of Epirus in Greece. This municipality was established in 2011 with the union of the former municipalities of Egnatia, Metsono, and Milia as applied with the administrative boundaries of the Kapodistrian Local Authorities. Metsovo is situated in the eastern part of the Regional Union of Ioannina, neighboring with the regions of Thessaly and Western Macedonia, and it extends in the slopes of Northern Pindos. In particular, Pindos is the biggest mountain range of Greece, stretching from the Greek–Albanian borders (northwest) to the Northern Peloponnese (southeast), roughly 160 km long and considered as the backbone of mainland Greece. Metsovo is listed as one of the most alpine municipalities of Greece, located at an elevation of 1160 m. The town of Metsovo is populated by 3469 residents, and it includes not only the administrative division of Metsovo but also the local communities of Anilio, Anthochori, and Votonosi.

In the mountainous area of Metsovo, which is the largest Vlach village of Greece, there are two very important museums, namely, the Folklore Museum housed in the mansion of Baron Michael Tositsa, and the Averoff Gallery, which has operated since 1988. The cultural heritage of the area includes architecture monuments, creative craftsmanship, the art of weaving and wood carving, a great embroidery history, folklore art, and traditional stone buildings, while its natural heritage is mainly characterized by an impressive mountainous bulge, a hydrographic network, and forests of a high density. Major important species of the flora include *Quercus sessiliflora, Acer platanoides, Acer pseudoplatanus, Ulmus montana Pinua nigra Erica carnea, Vacinium myrtillus, Quercus pubescens, Quercus conferta, Quercus cerris, Ostrya carpinifolia, Fraxinus ornus, Pinus leucodermis Abies borisii regis, Lilium heldreichii* and fauna including *Rupicarpa rupicarpa Lutra lutra, Ursus arctos,* and *Canis lupus.*

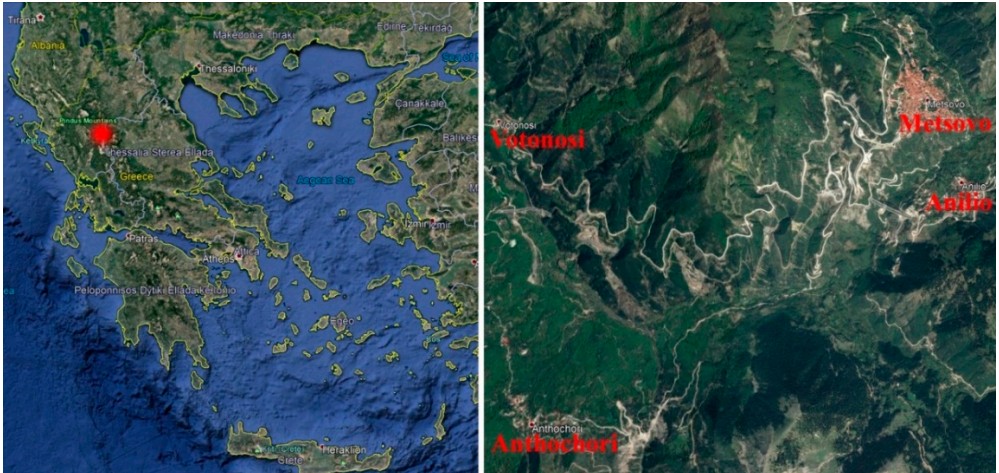

**Figure 1.** The location and map of the Municipality of Metsovo, Greece (Source: Google Maps).

*2.2. The Survey*

The population under study included all of the households in the Municipality of Metsovo. The applied sampling framework involved lists of domestic electricity consumers. The use of households is a familiar case of using teams instead of sample units. It is easier and more affordable [28]. Structured face-to-face interviews were conducted, and simple random sampling was used [29,30]. The average duration of the interview was 20 min. The survey was divided into five sections:

1. The demographic profile of the locals;
2. Residents' satisfaction with their quality of life in their region;
3. Evaluation of the existing infrastructures of tourism and forest recreation;
4. Evaluation of forest goods and services; and
5. Attitudes towards the State and private sectors' work on forests.

The data were collected in 2018 between the months of April to June. The interviews were conducted by a student who was well trained on the interviewing style, 17:00–19:00, on weekdays and on the weekend. The households were found randomly, using tables of random numbers. The personal interview was conducted with one family member per household. The response rate of the survey (98%) was very high. Participants had to be at least 18 years old due to legal constraints in Greece. If a member of a specific household was not found or refused to complete the questionnaire, we proceeded toward new sample units.

For the data collection, face-to-face interviews took place by means of a combination of close-ended questions and Likert-scale questions. The questionnaires included a wide range of topics in order to investigate their views on sectors of local growth, the impact of mountain tourism, the existing tourism and leisure activity infrastructures and projects, and possible problems in the forests, as well the role of Forest Service in traditional and alternative forest management.

*2.3. Research Method*

The population proportion, *p*, as well as the estimation of the standard error of the population, $s_p$, were calculated through the use of the formulas of simple random sampling. To determine the sample size, presampling was used, with a sample size of 50 individuals. The size of the sample for each variable was estimated on the basis of the formulas of simple random sampling (where *t* = 1.95 and *e* = 4.9%) [31]. In this case, the size of the sample was calculated as 400 individuals. Data collection took place in 2018 from April to June.

In the multivariables, reliability and factor analysis were applied. In particular, to find out the internal reliability of a questionnaire [32], i.e., if our data had the tendency to measure the same thing,

we used the coefficient (or Cronbach's reliability coefficient). A coefficient equal to or higher than 0.70 is considered as satisfactory [33], while one higher than 0.80 is considered as very satisfactory. In practice, reliability coefficients with values lower than 0.60 are also accepted in many cases [34].

Tests must be reliable in order to be useful. In fact, reliability does not suffice; it should also be valid, and this is checked through factor analysis [34]. Factor analysis is a statistical method that aims to discover the existence of common factors within a group of variables [35]. More specifically, principal component analysis was used in the case study, which is based on the spectral analysis of the variance (correlation) matrix. The selection of the number of factors is a dynamic process and presupposes the evaluation of the model in a repeated fashion. In this paper, we used the solution of two factors. We also conducted the rotation of the matrix principal components using the maximum variance rotation method by Kaiser [36,37].

Moreover, the hierarchical cluster analysis was applied to analyze data sets comprising multiple variables and to identify possible groupings of the data [31]. The Pearson correlation coefficient was used as the distance measure, and the full bond method, also known as the "nearest neighbor" method, was used to combine the cluster observations. The data were analyzed through the statistical package SPPS.

## 3. Results

### 3.1. The Demographic Profile of the Locals

The group under investigation was chosen to be the local residents and not the tourists. Indeed, after an elaborative examination of similar studies, namely, in the Island of Skiathos, in Dadia, and Evros Delta, Greece, it was observed that locals add more rigor in their assessment on tourism and/or leisure infrastructures mainly because these infrastructures are regarded as very important for the improvement of their life quality, and they are closely affiliated with activities that take place on a daily basis. In comparison with tourists, it was concluded that the latter are more lenient and assess tourism infrastructures in a more superficial way [24–26]. Additionally, if the state intends to introduce local support for tourism, tourism policy must be sensitive to the needs and interests of the locals, as community support to the locals is also engaged with local growth [38]. The demographic characteristics of the local people of Metsovo were recorded during the interviews that took place with the locals, and they are listed in Table 1. Apparently, there was an almost equal distribution of both sexes, with the majority having completed high school education (31.2%); they were primarily married (51.0%) with children (50.5%), and regarding their occupation, they were mainly freelancers (27%).

**Table 1.** Demographic characteristics of the locals.

| | | | | |
|---|---|---|---|---|
| **Gender** | **Male**<br>51.2% ($s_p$ = 0.0250) | **Female**<br>48.8% ($s_p$ = 0.0250) | | |
| **Age** | 18–30<br>35.5% ($s_p$ = 0.0240) | 31–40<br>21.0% ($s_p$ = 0.0204) | 41–50<br>23.0% ($s_p$ = 0.0211) | >50<br>20.5% ($s_p$ = 0.0202) |
| **Marital status** | **unmarried**<br>43.0% ($s_p$ = 0.0248) | **married**<br>51.0% ($s_p$ = 0.0250) | **divorced or widowed**<br>6.0% ($s_p$ = 0.0119) | |
| **Childhood**<br>**without children**<br>49.5% ($s_p$ = 0.0250) | **one child**<br>11.0% ($s_p$ = 0.0157) | **two children**<br>24.5% ($s_p$ = 0.0215) | **three children**<br>10.5% ($s_p$ = 0.0153) | **more than three**<br>4.5% ($s_p$ = 0.0104) |
| **Educational level** | **primary school**<br>13.8% ($s_p$ = 0.0172) | **Secondary school**<br>10.5% ($s_p$ = 0.0153) | **technical school**<br>15.0% ($s_p$ = 0.0179) | |
| | **High school**<br>31.2% ($s_p$ = 0.0232) | **technological ed.**<br>13.2% ($s_p$ = 0.0170) | **University**<br>16.2% ($s_p$ = 0.0185) | |
| **Profession**<br>**forest employees**<br>3.0% ($s_p$ = 0.0085) | **private employee**<br>16.5% ($s_p$ = 0.0186) | **public servants**<br>6.0% ($s_p$ = 0.0119) | **farmers \|**<br>**livestock farmers**<br>6.8% ($s_p$ = 0.0126) | **pensioners**<br>5.0% ($s_p$ = 0.0109) |
| **freelancers**<br>27.0% ($s_p$ = 0.0222) | **students**<br>15.0% ($s_p$ = 0.0179) | **housewives**<br>13.8% ($s_p$ = 0.0172) | **unemployed**<br>7.0% ($s_p$ = 0.0128) | |

### 3.2. Mountain Tourism Growth and Quality of Life

According to Sirakaya [38] and Roberts [39], it is obvious that the economic dimension of sustainable development is based on the satisfaction of materialistic human needs and includes the financial support of the local population and job creation. In line with this approach, it was important to investigate whether the locals were satisfied with their income and quality of life. Indeed, 41.8% of the locals claimed that they were satisfied and 36.8% less satisfied with their incomes, while their views of the quality of life offered in their area were more positive, with 50% considering that they were satisfied and 24.5% very satisfied (Figure 2). Unfortunately, only 26.8% ($s_p$ = 0.0222) of the locals believed that there are certain prerequisites for young people to be resident in the area, while 73.2% ($s_p$ = 0.0222) considered that these prerequisites are not yet established in Metsovo. Therefore, it should be noted that the central administration should facilitate decentralization motivations and trends by advancing the services and prospects provided in rural areas, something that may be achieved through the balance between the local and central administration interests [40].

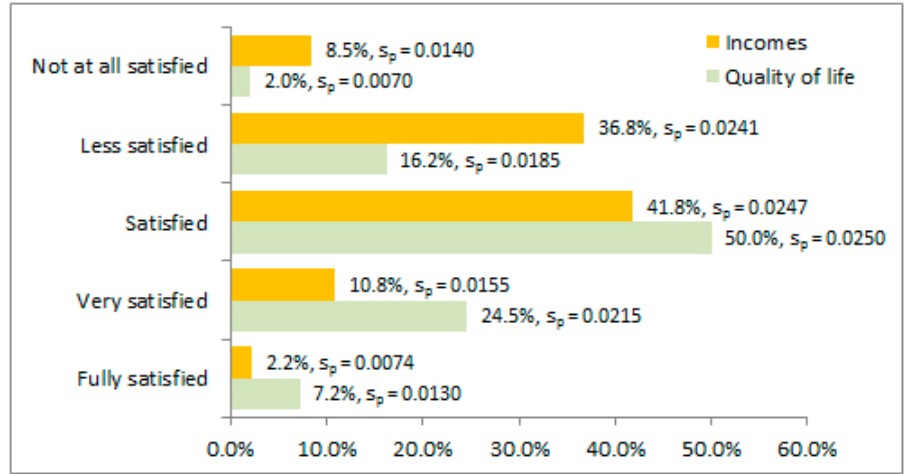

**Figure 2.** Locals' satisfaction with their perceived incomes and their quality of life.

Populations that reside in forest areas and rural areas depending on the natural resource-based industry are exclusively employed in primary production [41], while a significant contribution of their quite low incomes is based on forest and nonforest environmental incomes [42]. Agriculture, livestock farming, and forestry were traditionally and still remain interdependent activities characterizing the main economic activity sectors for the population of the semimountainous and mountainous areas of the Greece and the basis for their local growth [43,44]. In the case study, the locals assessed tourism (88%), livestock farming (62.2%), and forestry (36.8%) as the most important sectors for the local growth of the area, with less acceptance for the trade (18.2%), agricultural field (14%), industry (7.2%), cottage industry (4%), and construction sector (3.8%) (Figure 3). To this end, it is obvious that the residents of Metsovo are looking forward to opportunities for local growth deriving from the tourism industry, considering that the development of sustainable tourism acts as an alternative, providing opportunities for rural development, and at the same time motivates both locals and visitors for the preservation of the natural environment [45,46]. In particular, the locals' main occupation is focused on the sectors of livestock farming, cheese and wine-making, forestry, and the cottage industry, as well as folk art and the textile industry, while, in recent decades, there has been a rising trend on mountain tourism development in the area. Concerning the impact of this growth, they regard that mountain tourism development has a positive (26%) and very positive (29%) or even neutral (42.5%) effect on their lives, and a neutral one on the forest (74.8%) (Figure 4).

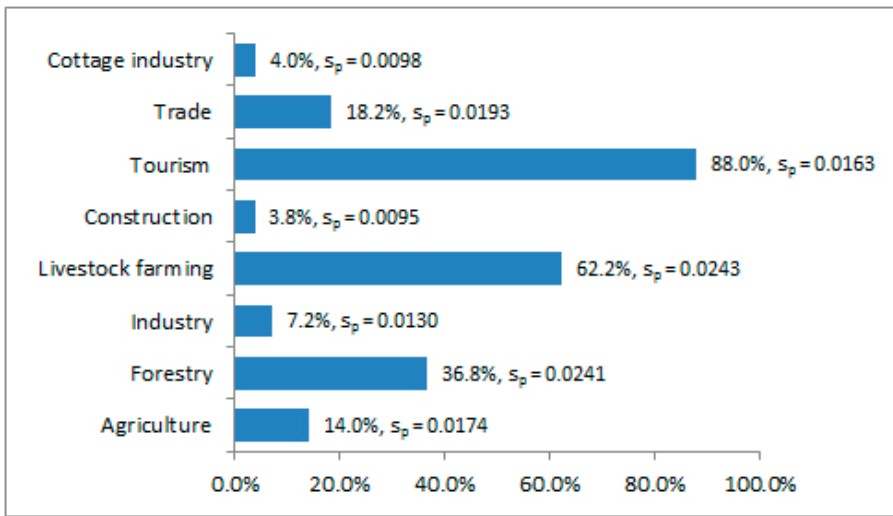

**Figure 3.** The importance of certain sectors to economic growth of Metsovo.

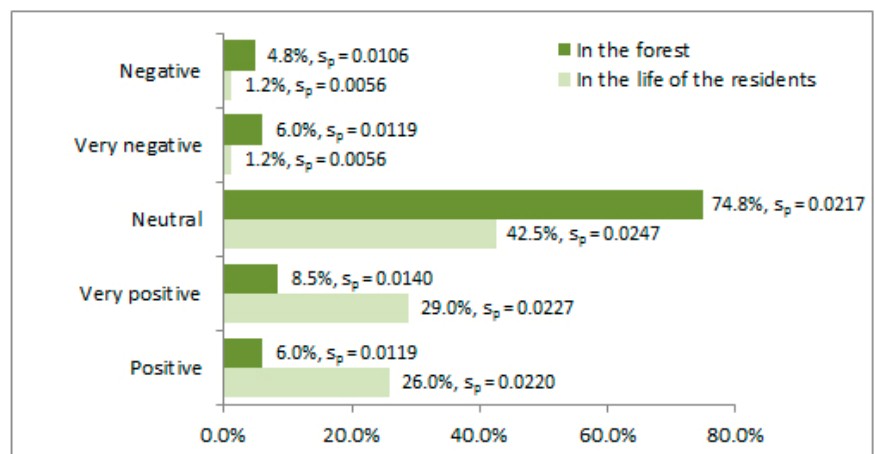

**Figure 4.** The impact of mountain tourism growth to locals' life and on the forest.

### 3.3. Evaluation of the Infrastractrural Development

The existence of infrastructures is an essential prerequisite for tackling strong competition in the tourism industry and for the development of a region, both nationally and internationally [21]. Transportation is considered as a fundamental precondition for tourists increasing their mobility and speed of move on the node of destination origin and as it is termed by Page and Ge [47], and it has a symbiotic relationship with the tourism concept. Therefore, the provision of a satisfactory level of road network connectivity is one of the most important factors affecting accessibility in mountain tourism [48], whereas accessibility problems and adequate transportation links constitute a common challenge for mountain areas aiming to attract visitors [49,50]. Means of transport and the road network are factors that determine the quality of the trip (Pettebone et al. 2011). In Metsovo, 46% of the locals assessed the road network as mediocre, 28% as good, and 16.2% as poor, and 39.2% assessed public transportation in their area as mediocre, 26.5% as good, and 21.2% poor (Table 2). Interestingly, they have a positive view of the hotel facilities and the rooms to rent in their area, as well as the ski resorts. Moreover, 49.8% rate the hotel facilities as very good and 42% as good, while the ski resorts are assessed by 65.5% as very good and 26.8% as good (Table 2). In the Municipality of Metsovo, there are three ski resorts, namely, Politses (altitude 1360–1620 m), Anilio (altitude 1680–1850 m), and Karakoli (altitude 1340–1520 m), all situated in a close proximity of less of than 10 km away from the center of Metsovo. According to the locals' views, cleanliness and waste management services are good (54.2%) and

mediocre (26%), while restaurants and recreation areas are rated as good (51.2%). Finally, residents regard the forest recreation infrastructure in their area as poor (29%) and very poor (34.5%) (Table 2).

**Table 2.** Assessment of the tourism infrastructures by locals.

| Tourism Infrastructure | | Very Good | Good | Mediocre | Poor | Very Poor |
|---|---|---|---|---|---|---|
| Road network | % | 5.8% | 28.0% | 46.0% | 16.2% | 4.0% |
| | $s_p$ | 0.0117 | 0.0225 | 0.0250 | 0.0185 | 0.0098 |
| Public transport (buses) | % | 1.5% | 26.5% | 39.2% | 21.2% | 11.5% |
| | $s_p$ | 0.0061 | 0.0221 | 0.0244 | 0.0205 | 0.0160 |
| Cleanliness and Waste Management | % | 9.0% | 54.2% | 26.0% | 8.5% | 2.2% |
| | $s_p$ | 0.0143 | 0.0249 | 0.0220 | 0.0140 | 0.0074 |
| Hotels | % | 49.8% | 42.0% | 6.5% | 1.2% | 0.5% |
| | $s_p$ | 0.0250 | 0.0247 | 0.0123 | 0.0056 | 0.0035 |
| Restaurants and Recreation areas | % | 24.5% | 51.2% | 20.2% | 2.8% | 1.2% |
| | $s_p$ | 0.0215 | 0.0250 | 0.0201 | 0.0082 | 0.0056 |
| Ski resorts | % | 65.5% | 26.8% | 5.8% | 1.0% | 1.0% |
| | $s_p$ | 0.0238 | 0.0222 | 0.0117 | 0.0050 | 0.0050 |
| Forest recreation infrastructure | % | 3.0% | 7.8% | 25.8% | 29.0% | 34.5% |
| | $s_p$ | 0.0085 | 0.0134 | 0.0219 | 0.0227 | 0.0238 |

Reliability analysis and factor analysis were applied to the variables concerning the evaluation of infrastructures. We applied reliability analysis to the above variables after completing all the necessary checks. The value of the reliability coefficient alpha was 0.642. This constitutes a strong indication that our data have the tendency to measure the same thing. In fact, this is also supported by the significantly high partial reliability coefficients alpha after the deletion of any variable, since even then, no increase of the reliability coefficient was observed. Additionally, before proceeding with the application of factor analysis, we conducted all the necessary checks. The value of the Keiser–Meyer–Olkin indicator was 0.694. Furthermore, Bartlett's test of sphericity rejected the null hypothesis that the correction table is unitary and that the partial correlation coefficients are low. The fact that the measures of sampling adequacy have high to very high values also supports the view that the factor analysis model is acceptable. The factors extracted are two. Table 3 reveals the loads that are the partial correlation factors of the seven variables, with each of the two factors resulting from the analysis. The higher the load of a variable in a factor, the more this factor is responsible for the total degree of fluctuation of the considered variable.

**Table 3.** Factor analysis loadings after rotation (bold numbers show the factor that belongs to each variable).

| Variables | Factor Loadings | |
|---|---|---|
| | 1 | 2 |
| Road network | 0.239 | **0.698** |
| Public transport (buses) | **0.667** | 0.039 |
| Cleanliness and Waste Management | **0.704** | −0.006 |
| Hotels | **0.727** | 0.042 |
| Restaurants and Recreation areas | **0.589** | 0.354 |
| Ski resorts | **0.633** | −0.149 |
| Forest recreation infrastructure | −0.292 | **0.770** |

The burdens given in bold show which variables are included in each factor

The first factor was termed "Tourism infrastructure: based on private initiative" as the group highly valued the infrastructures that are developed by investments of the private sector and are based on

entrepreneurship mindsets, while the second factor was called "Tourism infrastructure based on central administration". These findings are closely affiliated to the fact that the Greek government had been directly contributing to the promotion of tourism development since 1950; whereas at the beginning of the 1980s, the state turned to the encouragement and promotion of incentives for private investment in the tourism sector [51]. In the case study, it is obvious that the local people of Metsovo realize that tourism infrastructures developed by the central administration are inadequate in comparison to the ones created and managed by private initiatives. In addition, the tourism development policy implemented in Greece has been rather fragmented and was introduced without any strategical planning [52]. Indeed, according to Varvaressos [53], "simplistic empiricism", which reflected the prevailing concept of tourism policy actors, has proven to serve as another practical difficulty in designing and implementing new and state-of-the-art policy and strategies in the tourism sector.

To this end, the local people of Metsovo perceive that the development of mountain tourism in their area constitutes another paradigm of simplistic empiricism, which implies the existence of leisure infrastructures, distributed in the area in order to support leisure activities addressed at tourists. In this respect, and based on the fact that forest recreation is a special form of outdoor recreation, forest activities, facilities, and services provided in mountain tourism should be established in the same concept to meet the visitors' needs and expectations and at the same time as preserving forest resources [22]. Nevertheless, in the case of Metsovo, the forest facilities and infrastructures were evaluated to be below average, except for the paths and water provision (Table 4). On the other hand, the creation and maintenance of paths should be considered as a major priority for the area as they were highly related with income-generating activities, such as trekking and mountaineering, while cycling was also regarded as a popular activity related to mountain adventure tourism [54]. However, as regards other important activities, it was stated by the locals that there is a lack of first aid spots (64.8%), restrooms (64.2%), information centers (50.5%), sports areas (49.8%), and parking areas (39.5%).

**Table 4.** Assessment of the recreation infrastructures by the locals.

| Recreation Infrastructure | | Very Good | Good | Mediocre | Poor | Very Poor | Not Exist |
|---|---|---|---|---|---|---|---|
| Water provision | % | 11.5% | 26.0% | 31.0% | 15.2% | 11.0% | 5.2% |
| | $s_P$ | 0.0160 | 0.0220 | 0.0232 | 0.0180 | 0.0157 | 0.0112 |
| Sports areas | % | 0.8% | 11.2% | 17.8% | 13.5% | 7.0% | 49.8% |
| | $s_P$ | 0.0043 | 0.0158 | 0.0191 | 0.0171 | 0.0128 | 0.0250 |
| Paths | % | 9.2% | 32.5% | 33.0% | 12.8% | 4.5% | 8.0% |
| | $s_P$ | 0.0145 | 0.0234 | 0.0236 | 0.0167 | 0.0104 | 0.0136 |
| Waste bins and Cleanliness | % | 1.2% | 18.8% | 29.2% | 17.8% | 19.8% | 13.2% |
| | $s_P$ | 0.0056 | 0.0195 | 0.0228 | 0.0191 | 0.0199 | 0.0170 |
| Restrooms | % | 0.2% | 2.5% | 9.2% | 15.8% | 8.0% | 64.2% |
| | $s_P$ | 0.0025 | 0.0078 | 0.0145 | 0.0182 | 0.0136 | 0.0240 |
| Infrastructures for leisure activities | % | 2.8% | 14.2% | 23.8% | 17.0% | 29.0% | 13.2% |
| | $s_P$ | 0.0082 | 0.0175 | 0.0213 | 0.0188 | 0.0227 | 0.0170 |
| Parking areas | % | 1.0% | 8.8% | 21.5% | 13.0% | 16.2% | 39.5% |
| | $s_P$ | 0.0050 | 0.0141 | 0.0206 | 0.0168 | 0.0185 | 0.0245 |
| First aid spots | % | 0.2% | 5.0% | 15.0% | 8.8% | 6.2% | 64.8% |
| | $s_P$ | 0.0025 | 0.0109 | 0.0179 | 0.0141 | 0.0121 | 0.0239 |
| Information centers | % | | 4.2% | 19.8% | 16.0% | 9.5% | 50.5% |
| | $s_P$ | | 0.0101 | 0.0199 | 0.0184 | 0.0147 | 0.0250 |
| Informational signs | % | 2.5% | 17.2% | 32.2% | 14.2% | 14.8% | 19.0% |
| | $s_P$ | 0.0078 | 0.0189 | 0.0234 | 0.0175 | 0.0178 | 0.0196 |

Reliability and factor analysis were applied to the variables referring to the evaluation of the existing infrastructures located in the recreation areas, after completing all the necessary checks for their acceptance. The value of the reliability coefficient alpha was 0.876, while for the factor analysis, the value of the Keiser–Meyer–Olkin indicator was 0.868. The factors extracted were three. Table 5

reveals the loads that are the partial correlation factors of the 10 variables with each of the three factors resulting from the analysis. The first factor can be named as "Primary recreational infrastructures". The second factor can be called "Information and accessibility infrastructures" and the third one "Trekking infrastructures".

**Table 5.** Factor analysis loadings after rotation (bold numbers show the factor that belongs to each variable).

| Variables | Factor Loadings | | |
|---|---|---|---|
| | **1** | **2** | **3** |
| Water provision | 0.152 | 0.183 | **0.832** |
| Sports areas | **0.680** | 0.419 | 0.273 |
| Paths | 0.175 | 0.145 | **0.850** |
| Waste bins & Cleanliness | **0.739** | −0.217 | 0.383 |
| Restrooms | **0.825** | 0.344 | 0.055 |
| Infrastructures for leisure activities (benches etc.) | **0.539** | 0.492 | 0.247 |
| Parking areas | 0.241 | **0.708** | 0.263 |
| First aid spot | **0.762** | 0.399 | 0.065 |
| Information centers | 0.358 | **0.732** | 0.168 |
| Informational signs | 0.075 | **0.873** | 0.027 |

The burdens given in bold show which variables are included in each factor.

Accordingly, as depicted in Table 6, the locals claim to be less or not at all satisfied with the infrastructures for leisure activities, such as excursion and campsites; environmental information centers; and paths for the observation of nature, developed by the Forest Service, it being the major responsible service for forest management issues in their area. Reliability and factor analysis were applied to the variables referring to the evaluation of infrastructures for leisure activities developed by the Forest Service, after completing all the necessary checks for their acceptance. The value of the reliability coefficient alpha was 0.880, while for the factor analysis, the value of the Keiser–Meyer–Olkin indicator was 0.810. The four variables belong to the same factor that is termed as "Satisfaction by the Forest Service". The situation turns out to be more challenging due to the fact that their views about the Forest Service were neutral (31%, $s_p$ = 0.0232), very positive (25.8%, $s_p$ = 0.0219), and positive (24.2%, $s_p$ = 0.0215) while a small percentage of 12.5% ($s_p$ = 0.0166) were very negative and 6.5% were negative ($s_p$ = 0.0123).

**Table 6.** Satisfaction level for the recreation projects supervised by the Forest Service.

| Project | | Absolutely Satisfied. | Very Satisfied. | Satisfied | Less Satisfied. | Not at All Satisfied |
|---|---|---|---|---|---|---|
| Excursion areas | % | 1.0% | 3.2% | 12.5% | 26.0% | 57.2% |
| | $s_p$ | 0.0050 | 0.0089 | 0.0166 | 0.0220 | 0.0248 |
| Camping sites | % | 0.2% | 2.0% | 4.2% | 24.2% | 69.2% |
| | $s_p$ | 0.0025 | 0.0070 | 0.0101 | 0.0215 | 0.0231 |
| Nature trails | % | 1.5% | 6.8% | 15.8% | 27.5% | 48.5% |
| | $s_p$ | 0.0061 | 0.0126 | 0.0182 | 0.0224 | 0.0250 |
| Environmental information centers | % | 0,2% | 2.8% | 9.8% | 26.0% | 61.2% |
| | $s_p$ | 0.0025 | 0.0082 | 0.0149 | 0.0220 | 0.0244 |

*3.4. Forest-Based Assets*

The value attributed to forest goods and services by the locals as well as its existing problems highlight the importance the locals ascribe to the forest for the development of their area. As very important forest goods and services, the locals assessed the production of oxygen (84.8%), the fact that it cleans the atmosphere from pollution (67.8%), that it conserves wildlife (62.5%), and also that it

produces wood products (61.5%), meaning the goods and services received due to traditional forest management. The goods and services that follow were prevention from floods (57.5% very important and 30.8% important) and soil erosion (51.5% very important and 33.2% important), while also the fact that the forest increases water reserves (44% very important and 42.2% important), which is listed in the forest goods and services produced from the management of forest hydrology (Table 7). Finally, the aspect of recreation opportunities was characterized as moderate by 32% of locals, as important by 29.8%, and as very important by 23.8%, while the creation of employment opportunities was claimed to be slightly important by 40.8% and moderately important by 34.2%. The value of the natural environment in attracting tourists is of utmost importance, as nature and cultural heritage represent a competitive advantage for many areas in Greece and worldwide. However, it appears that the real value of the natural environment in the broader area has not yet been attributed to its real extent in the locals' mindsets.

**Table 7.** Assessment of local forest ecosystem goods and services by the locals.

| Goods and Services | | Very Important | Important | Moderately Important | Slightly Important | Not Important |
|---|---|---|---|---|---|---|
| Produces wood products | % | 61.5% | 26.8% | 10.0% | 1.5% | 0.2% |
| | $s_p$ | 0.0244 | 0.0222 | 0.0150 | 0.0061 | 0.0025 |
| Provides recreation opportunities | % | 23.8% | 29.8% | 32.0% | 11.5% | 3.0% |
| | $s_p$ | 0.0213 | 0.0229 | 0.0234 | 0.0160 | 0.0085 |
| Produces oxygen | % | 84.8% | 12.0% | 2.5% | 0.5% | 0.2% |
| | $s_p$ | 0.0180 | 0.0163 | 0.0078 | 0.0035 | 0.0025 |
| Increases water reserves | % | 44.0% | 42.2% | 12.0% | 1.8% | |
| | $s_p$ | 0.0249 | 0.0247 | 0.0163 | 0.0066 | |
| Prevents soil erosion | % | 51.5% | 33.2% | 13.5% | 1.5% | 0.2% |
| | $s_p$ | 0.0250 | 0.0236 | 0.0171 | 0.0061 | 0.0025 |
| Prevents from floods | % | 57.5% | 30.8% | 10.5% | 0.8% | 0.5% |
| | $s_p$ | 0.0247 | 0.0231 | 0.0153 | 0.0043 | 0.0035 |
| Creates employment opportunities | % | 6.0% | 17.2% | 34.2% | 40.8% | 1.8% |
| | $s_p$ | 0.0119 | 0.0189 | 0.0238 | 0.0247 | 0.0066 |
| Conserves wildlife | % | 62.5% | 21.8% | 12.8% | 1.8% | 1.2% |
| | $s_p$ | 0.0242 | 0.0207 | 0.0167 | 0.0066 | 0.0056 |
| Cleans the atmosphere from pollution | % | 67.8% | 24.0% | 6.2% | 1.5% | 0.5% |
| | $s_p$ | 0.0234 | 0.0214 | 0.0121 | 0.0061 | 0.0035 |

Reliability and factor analysis were then applied to the variables referring to the evaluation of the goods and services provided by the forest of the area, after completing all the necessary checks for their acceptance. The value of the reliability coefficient alpha was 0.736, while for the factor analysis, the value of the Keiser–Meyer–Olkin indicator was 0.784. The factors extracted were three. Table 8 introduces the loads that are the partial correlation factors of the nine variables with each of the three factors resulting from the analysis. The first factor could be named "Goods and services coming from the forest water management", the second factor "Traditional forest goods and services", while the third factor is "Opportunities for recreation and tourism development".

Following that, the locals were asked to assess the existing problems in the forests of their area. To this point, overgrazing by herds of animals was highlighted as the most important existing problem (40.8% average and 24.8% slight), followed by illegal hunting (33% moderately important and 21.8% slightly important) and forest land loss due to human encroachment (28.5% moderately and 31.2% slightly important) (Table 9). Forest fires were not considered as an important problem (67.5%), nor were insect and disease outbreaks (67%) or the large number of visitors (51%). Regarding the security and policing of the forests in their area, the locals stated being satisfied (42%, $s_p$ = 0.0247) and less satisfied (31.5%, $s_p$ = 0.0233), with 13.2% ($s_p$ = 0.0170) claiming to be not at all satisfied, 9.8% ($s_p$ = 0.0149) were very satisfied, and 3.5% ($s_p$ = 0.0092) were absolutely satisfied.

**Table 8.** Factor analysis loadings after rotation (bold numbers show the factor that belongs to each variable).

| Variables | Factor Loadings | | |
|---|---|---|---|
| | 1 | 2 | 3 |
| Produces wood products | −0.154 | **0.815** | 0.080 |
| Provides recreation opportunities | 0.149 | 0.113 | **0.764** |
| Produces oxygen | 0.307 | **0.651** | 0.051 |
| Increases water reserves | **0.680** | 0.225 | 0.325 |
| Prevents soil erosion | **0.858** | 0.171 | 0.075 |
| Prevents from floods | **0.843** | 0.137 | 0.086 |
| Creates employment opportunities | 0.085 | −0.073 | **0.815** |
| Conserves wildlife | 0.355 | **0.583** | −0.157 |
| Cleans the atmosphere from pollution | 0.369 | **0.695** | 0.079 |

The burdens given in bold show which variables are included in each factor.

**Table 9.** Assessment of the existing problems in the forests of the area.

| Kind of Ploblem | | Not Important | Slightly Important | Moderately Important | Important | Very Important |
|---|---|---|---|---|---|---|
| Illegal Hunting | % | 20.0% | 21.8% | 33.0% | 19.5% | 5.8% |
| | $s_P$ | 0.0200 | 0.0207 | 0.0235 | 0.0198 | 0.0117 |
| Overgrazing by herds of animals | % | 11.2% | 24.8% | 40.8% | 18.0% | 5.2% |
| | $s_P$ | 0.0158 | 0.0216 | 0.0246 | 0.0192 | 0.0112 |
| Forest lands loss due to human encroachment | % | 23.8% | 31.2% | 28.5% | 12.5% | 4.0% |
| | $s_P$ | 0.0213 | 0.0232 | 0.0226 | 0.0166 | 0.0098 |
| Forest fires | % | 67.5% | 21.2% | 7.8% | 2.0% | 1.5% |
| | $s_P$ | 0.0234 | 0.0205 | 0.0134 | 0.0070 | 0.0061 |
| Insect and disease outbreaks | % | 67.0% | 20.2% | 9.5% | 2.2% | 1.0% |
| | $s_P$ | 0.0235 | 0.0208 | 0.0147 | 0.0074 | 0.0050 |
| Large number of visitors | % | 51.0% | 20.8% | 18.5% | 8.0% | 1.8% |
| | $s_P$ | 0.0250 | 0.0208 | 0.0194 | 0.0136 | 0.0066 |

Reliability and factor analysis were applied to the variables referring to the existing problems in the forests of the area, after completing all the necessary checks for their acceptance. The value of the reliability coefficient alpha was 0.811, while for the factor analysis, the value of the Keiser–Meyer–Olkin indicator was 0.804. The factors extracted were two. Table 10 reveals the loads that are the partial correlation factors of the six variables, with each of the two factors resulting from the analysis. The first factor could be titled "Primary problems of the forests" and the second factor is "Problems arising from delinquent behavior".

**Table 10.** Factor analysis loadings after rotation (bold numbers show the factor that belong to each variable).

| Variables | Factor Loadings | |
|---|---|---|
| | 1 | 2 |
| Illegal Hunting | 0.174 | **0.794** |
| Overgrazing by herds of animals | 0.117 | **0.835** |
| Forest lands loss due to human encroachment | 0.416 | **0.719** |
| Forest fires | **0.860** | 0.163 |
| Insect and disease outbreaks | **0.865** | 0.166 |
| Large number of visitors | **0.728** | 0.291 |

The burdens given in bold show which variables are included in each factor.

With the hierarchical cluster analysis, we grouped the factors from the above five factor analyses. As shown in the dendrogram of the factors' correlation (Figure 5), information and accessibility infrastructures in the recreational area are closely affiliated with satisfaction from the Forest Service, based on central administration and recreational tourism development opportunities. In a major distance, the primary recreational and trekking infrastructures factors are connected, forming the first cluster titled "Association of central administration and forest recreation". The second cluster can be termed as "Association of the local communities with tourism development", in which infrastructures based on private initiatives are related with traditional forest goods and services, forest problems, and forest water management, along with the delinquent behavior of citizens. Indeed, local communities perceive the forest and forest lands as a natural resource—a landscape that supports tourism development. Correspondingly, the relationships between forests and society relate to the benefits the local community receives from it and, conversely, the pressure the forest receives from it. Not to mention that forest recreation is a complimentary good provided by the state, which is associated with non-economic benefits in the locals' concepts. Indeed, it is listed as one of the major pillars for innovative forest management in Greece, aiming at the enlargement of forest economics in new horizons, including both nonwood products and the alternative services of forest recreation and mountain tourism [55].

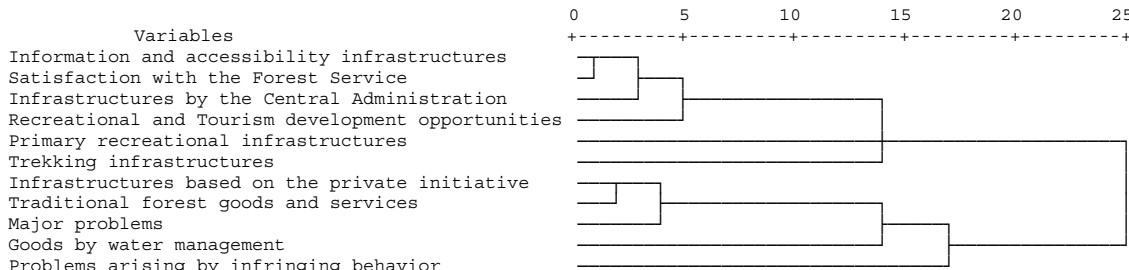

**Figure 5.** The impact of mountain tourism growth on the locals' life and on the forest.

## 4. Discussion

The major goal of Greek forestry policy has always been to meet the needs of mountainous populations through the sustainable management of forest ecosystems. However, their nonwood functions have been set aside, reflecting the effects of the depopulation and abandonment of rural areas [56]. In the Metsovo region, livestock farming and forestry have been recognized as the main occupations of the local people. However, the locals perceive that tourism (88%) may be the most important development sector in this area, with livestock farming (62.2%) and forestry (36.8%) following after, whereas the substantial span of mountain areas in Greece, which is unexploited to a large extent, could serve as the vehicle for local growth and sustainable development for forest gateway communities through the growth of mountain tourism [57], accompanied by the development of the respective infrastructures.

Forest recreation facilities and improvement of accessibility are considered of high priority for the development and efficient management of mountain tourism [55] as the existence of tourism infrastructures constitutes a comparative advantage for every region that is expected to succeed in tourism development. In the area of Metsovo, ski resorts and hotel facilities receive the highest rating, followed by restaurants and recreation areas, cleanliness and waste management, road networks, and public transport (buses), and forest recreation infrastructures are the lowest-rated aspect, showing the need for the establishment and maintenance of outdoor public infrastructures. In addition, it is evident that the local people of Metsovo realize that tourism infrastructures based on private initiatives have been developed significantly in contrast to those that should have been developed by the central administration. To this point, it should be noted that due to certain policies implemented in Greece, which aim to encourage private investment in the tourism sector, the country lacks an integrated

plan that would enhance and promote the various and special existing mountain and nature-based tourist destinations distinguished by their unique comparative advantages and remarkable features. For instance, in the mountainous Epirus region, there are also sacred forests that differ greatly from one another in terms of size, forest type, and management practices [58].

Concerning the ski resorts, these are attractive leisure activities for the visitors of Metsovo during specific months of the year. However, only the hiking infrastructures received positive criticism from the locals, while the information and accessibility infrastructures as well as the infrastructures engaged with cleanliness and recreation were assessed as mediocre. As for the other recreational infrastructures, such as first aid spots (64.8%), restrooms (64.2%), information centers (50.5%), sports areas (49.8%), and parking areas (39.5%), the locals declared that they actually do not exist. In particular, these recreational projects should be implemented by the Forest Service. The Forest Service should also try to promote the forests of Metsovo by increasing their value through their use as recreational areas. Visitor attractions, such as excursions and campsites, nature observation trails paths, and environmental information centers, should be created within the forests of the area. Unfortunately, there is a lack of these very important infrastructures, which leads to persistent disapproval of the local people as they state being less or not at all satisfied with the infrastructures developed by the Forest Service. In spite of this general disapproval, it is remarkable that the locals report a quite positive to neutral attitude toward the Forest Service, indicating that its other activities of traditional forestry in the area are considered effective. Not to mention that there is also a need for the improvement of the road network as transportation provisions, such as the road network, in terms of density, accessibility, quality, and maintenance, which is conceptualized as a crucial factor for tourism activity [47].

Moreover, the value that residents attribute to the goods and services provided by the forest also indirectly reflects the management model they visualize for these assets. As very important forest goods and services, the ones deriving from traditional forest management were evaluated (it produces oxygen, cleans the air from pollution, is a wildlife refuge, and produces wood products), followed by water management (flood protection, protection from erosion, and increase in water reserves), while lower ratings were received by the Forest Service to improve recreational opportunities on forest areas and tourism development (forest recreation and employment opportunities). However, the forest management policy should also take into consideration that the traditional management of forests has significant impacts on the quality of nature-based tourism landscapes and thus should thoroughly examine the potential tradeoffs arising from tourism utilization [24]. In particular, it should be conceptualized that all valuable goods and services of the forest ecosystems could serve as the tool for locals in understanding their relationship with nature and integrate mountain tourism development as a vital component of this bound [59].

On the other hand, significant existing problems of the forests in Metsovo are mentioned to be those caused by delinquent behavior (overgrazing of animals, illegal hunting, and forest land loss due to human encroachment), while not significant problems are regarded by the locals the main problems of forests (forest fires, insect and disease outbreaks, and the large number of visitors). In fact, regarding security issues and policing of the forests in their area, the locals state they are satisfied and less satisfied (42% and 31.5%, respectively). With the aim of the hierarchical cluster analysis, we grouped the factors of the five factor analyses into two clusters: "Association of central administration and forest recreation" and "Association of the local communities with tourism development". Therefore, as already mentioned, the locals have expectations from the central administration, and namely, through the Forest Service, to promote forest recreation in the mountainous area of Metsovo, while they strongly believe that the existing tourism development of their area (Metsovo) was not effectively associated with the recreational opportunities the mountainous and forest ecosystems really provide.

## 5. Conclusions

A comparatively significant young population of 18 to 40-year-olds resides in the area of Metsovo, one of the most alpine regions of Greece encompassing all the challenges of living in a remote and

mountainous area with certain deficiencies and provisions and also experiencing severe climate conditions and abandonment or depopulation schemes. The existing population, having already invested in the development of leisure infrastructures and services, anticipates support and mobilization by the central authorities and especially by the Forest Service, in order to develop, properly maintain, and expand critical forest recreation infrastructures that play a vital role in mountain tourism.

Indeed, certain benefits for local growth have been acknowledged by the locals of Metsovo, but the very low impact on the natural environment as a consequence of tourism was also stated. This reveals that the locals consider that there is still room for further extension in the tourism sector in their area; this could be achieved primarily with the contribution of the Forest Service and with the intervention of the municipality and the established foundations, in order to activate and exploit all possible means asserting financial and technical support by the government for the development of forest recreational infrastructures, such as paths and marked paths, paved walkways, forest roads, bikeways, signs, benches, kiosks, information centers, grilling amenities, water leisure activities, and other state-of-the-art outdoor facilities.

They are several case studies worldwide providing evident best practices for successfully taking advantage of forest recreation opportunities. The case of Ekeberg Park, a publicly-owned area in Oslo, Norway, includes a state-of-the-art sculpture and national heritage park offering innovative forest recreation amenities and an impressive international artwork collection, which is placed in the woods, offering a combination of art, history, and nature of the last 130 years of European art history [60]. In the same line and taking into account that the broader area of Metsovo is far from its unique natural heritage assets but also has a very significant cultural heritage and that two foundations, Tositsa and Averoff, that have appreciably supported the development of the area, the potentials for creating new paths for local growth and the promotion of both cultural and natural heritage should be appreciated and exploited by the local, regional, and central authorities.

Finally, local people seem to be ready to embrace such efforts towards strengthening the mountain tourism industry in Metsovo, as they have already appraised the multidimensional forest value of the area, and they believe that the existent recreation infrastructures and services have not yet negatively affected the natural environment. They also share the view that traditional forest management implemented so far by the Forest Service is moving in the right direction, whereas they identify the forest recreational potential in their area, which needs to be enhanced by the central administration and locally by the Forest Service.

**Author Contributions:** S.T. designed the research as well as the questionnaire used for the data collection and supervised the project progress. I.P. conducted the fieldwork and data collection. V.A. and P.K. conducted the data analysis and wrote the paper in close cooperation with S.T. and I.P. All authors contributed to the review of the paper and approved the final manuscript submitted for publication.

**Funding:** This research received no external funding.

**Conflicts of Interest:** The authors declare no conflict of interest.

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
