# Peer review of "The Growth of Mountain Tourism in a Traditional Forest Area of Greece"

_forests, doi:10.3390/f10111022_

Round 1
Reviewer 1 Report
General comments for the authors:
It seems like there is a really good paper in here somewhere, but it is currently lost in poor organization and awkward English language wording. Some of the sentences can be re-worded as I’ve indicated in the line-by-line comments below, but other portions of the paper will need significant attention in terms of the way that the paragraphs are structured and what information is presented. Rather than going through and making numerous small recommendations, I will offer a couple of general organizational elements for you to consider.
1) Literature review. First, clearly establish WHAT the research on mountainous forested areas says about conservation and management. Then explain how the Greek Forest Service has or hasn’t met these needs.
2) Methods: The study site description is adequate, but the maps could be improved for legibility. The test is currently too small to read. The methods should be clearly described, without confusing the reader on the Factor and Correlation analysis until you have explained the survey instrument and details such as whether it was mailed, what the questions were, and other considerations that went into the data collection stage.
The research method section (2.2) should be significantly overhauled to clearly indicate what was done, including the specific techniques used to collect and analyze the data. As it currently stands, the section is a bit confusing and does not use precise terminology or clear descriptions. For example, in line 144 the statement seems to portray a confusion between sampling techniques and a formula or model used to interpret the data.
3) The results should clearly state the findings, and avoid saying things about why tourists were not selected, for example. This information should go into the study design portion.
4) I envision that some of the local managers will be interested in the findings. How can you anticipate or address some of their concerns in the discussion?
For now, this paper isn’t ready for publication but perhaps with some significant revisions and heavy re-writing (including English language proofing), it will become a nice piece of scholarship on mountain tourism. I’d focus on what your data say about tourism in relation to other industries and what specific infrastructure would be needed to improve overall experiences.
Line-by-line comments for the authors:
17-20 This sentence is a bit unwieldy and confusing. Would suggest separating out the portions that describe the Metsovo area from the ways that the locals’ views were investigated. This might need to be three separate sentences.
28-30: In this final sentence I am unsure how the traditional forest management is “appraised,” and why that is different than identifying the recreational potential. It seems that perhaps they could go together and not stand in opposition from one another.
35: Remove the “a”
37: Use “With regards to” instead of “as”
41-42: wording is a bit awkward here
47-49: Is there an established set of terminology for “hard” and “soft,” or could these be rephrased as difficult vs. low-intensity?
56: Should be “extent”
97: Could likely eliminate the terms “Actually, indicative” as they don’t help clarify anything about the findings.
132: What is a “manifold relief?” This is unclear from the list of regional assets.
153: Would be good to know what the reliability coefficient was in this study.
201: Should be spelled “traditionally”
202: Should be spelled “characterizing”
220: this line should begin with a “the”
Line 226-9: These percentages could be presented in a more streamlined fashion, for example, to say that % residents assess the road network as mediocre. Currently, the results read in a slightly difference word order. Just by switching the word order, this would make more sense to the reader.
243: Substitute more accurate terminology for “data,” as it seems you are getting at study design or survey instrument language.
304/5: Spell-check the word “positive”
337: Clarify the “overfeeding by herds of animals” statement. As it reads now, not sure what this means.
Reviewer 2 Report
Specific comments:
There is a lot of very long sentences (more than 5 or 6 lines). Scientific writing need to be short, direct to the point. Bring your readers to easily understand your point. See examples: lines 73 to 79, 80 to 85.
Figures 1, 2, 3 and 4 are very week quality.
Research method section need some specific information:
How many people in the population? How did you recruit people? Did your sample correspond to the population?Line 174: a dot after daily basis, not ;
Table 1: remove the line after Gender
Lines 198-200: What about your sample? Do you have any idea of their employment sectors?
Line 205: Why do you use the term acceptance? The question was not about acceptance, but perception of importance, right?
Line 206: add an and between (4%) and the construction sector
Line 210: Miss a dot after [40,41]
Line 211: How could you extrapolate, from your results, that residents of Metsovo realize that sustainable development is achieved in their area? There is no question/answer about that topic.
Line 221: considered instead of considererd
Line 225: accessibility instead of accecibility, transportation instead of transportaion
Line 232: assessed instead of assessd
Table 2 and others: you never talk about the Sp. How do you interpret this indicator? What information it gives about your results?
Line 240: concerning instead of cocerning
Line 297: What do you mean by etc. Are there more topics not shown? If yes, why didn't show them?
Line 318: I think the 32% need to be associated with the moderately important category.
The discussion section is very poor. Most of the paragraphs are almost the same information already presented in the results section. There is no real interpretation and no openness to the usefulness of these results.
What is the point to all your Factor analysis loadings? And your final figure 4? Are you able to group the local's views and divide the population into those groups? Is there enough concentration into one point of view so you could make specifics recommendations?
Reviewer 3 Report
A very interesting manuscript, concerning not only forest tourism and recreation management but also the development of tourism in general in mountain areas. The Introduction contains a very well described background to the research. The authors presented in detail the reasons for undertaking the research. The purpose of the study was clearly highlighted as "examine and interpret the locals’ views in the mountain area of Metsovo, regarding the evaluation of different factors that are able to support and encourage the growth of mountain tourism”. For better legibility of the text it would be good to define a research problem or to adopt research hypotheses. Materials and Methods: 2.1. "Study Area" - contains basic data on physical and geographical conditions. I am missing here the characteristics of forests, what area they occupy, what is their structure (age, species composition), the structure of properties?. 2.2. "Research Method" - The authors paid a lot of attention to the principles of statistical analysis. It is worthwhile to supplement the description of the method with more detailed data on random sampling. Was the survey conducted at home, in the forest, in what conditions? There is also no reflection on the scientific limitations for the conducted research. Could the place and the way of conducting the research have influenced the results? Results: 3.1. Demographic profile of the respondents - clearly defined. 3.2. "Mountain tourism growth and quality of life…” - this subsection is very voluminous. Maybe the subchapter on for example "forest recreation infrastructure" had to be separated by 269 lines each. What about educational infrastructure in forests? - why was this aspect not taken into account by the authors? It seems to me that Tables 7 and 8 go beyond the field of work. It is a pity that there is no reference in the text to the demographic profile of the respondents. Were there any differences in the assessment of tourism and recreation infrastructure according to gender, age, etc.? Did the authors analyse it? Table 2 lists the assessed elements of tourism infrastructure. However, it is not clear what was the subject of the evaluation. For example, hotels were assessed on a "very good - very poor" scale, but what exactly was the subject of the assessment: the number of hotels in this area, the standard of hotels, their availability? Similarly with restaurants and ski resort (price of services, accessibility, condition of infrastructure)? Discussion: In fact, many of the elements typical for discussion are already in the Results chapter. I suggest that you discuss the results in the "Results" chapter and move the comment to the "Discussion" section (e.g. 259-269, 320-322). Conclusions: in my opinion, a well-presented summary of the deliberations